# The Neuromelanin Paradox and Its Dual Role in Oxidative Stress and Neurodegeneration

**DOI:** 10.3390/antiox10010124

**Published:** 2021-01-16

**Authors:** Alexandra Moreno-García, Alejandra Kun, Miguel Calero, Olga Calero

**Affiliations:** 1Chronic Disease Programme, Instituto de Salud Carlos III, 28029 Madrid, Spain; almgarcia@isciii.es; 2Centro de Investigación Biomédica en Red sobre Enfermedades Neurodegenerativas, 28031 Madrid, Spain; 3Biochemistry Section, Science School, Universidad de la República, 11400 Montevideo, Uruguay; akun@fcien.edu.uy; 4Protein and Nucleic Acids Department, Instituto de Investigaciones Biológicas Clemente Estable, 11600 Montevideo, Uruguay; 5Queen Sofia Foundation Alzheimer Center, Alzheimer Disease Research Unit, CIEN Foundation, 28031 Madrid, Spain

**Keywords:** reactive oxygen species (ROS), neuromelanin (NM), oxidative stress, neurodegeneration, immune response

## Abstract

Aging is associated with an increasing dysfunction of key brain homeostasis mechanisms and represents the main risk factor across most neurodegenerative disorders. However, the degree of dysregulation and the affectation of specific pathways set apart normal aging from neurodegenerative disorders. In particular, the neuronal metabolism of catecholaminergic neurotransmitters appears to be a specifically sensitive pathway that is affected in different neurodegenerations. In humans, catecholaminergic neurons are characterized by an age-related accumulation of neuromelanin (NM), rendering the soma of the neurons black. This intracellular NM appears to serve as a very efficient quencher for toxic molecules. However, when a neuron degenerates, NM is released together with its load (many undegraded cellular components, transition metals, lipids, xenobiotics) contributing to initiate and worsen an eventual immune response, exacerbating the oxidative stress, ultimately leading to the neurodegenerative process. This review focuses on the analysis of the role of NM in normal aging and neurodegeneration related to its capabilities as an antioxidant and scavenging of harmful molecules, versus its involvement in oxidative stress and aberrant immune response, depending on NM saturation state and its extracellular release.

## 1. Role of Metals and Oxidative Stress in the Local Origin of Neurodegenerative Diseases

Neurodegenerative diseases are characterized by progressive cognitive dysfunction and loss of neurons and synapses. Neurodegeneration is a process extremely complex, which involves a multifactorial imbalance with the simultaneous interplay of numerous processes operating at different levels of functional organization. In this dynamic context, neurodegeneration may be considered a pathological brain-aging process, triggered by alterations on distinct molecular pathways, genetic predisposition, and environmental toxic exposure. Therefore, aging is by far the greatest risk factor for highly prevalent neurodegenerative disorders [1].

A key feature for most neurodegenerative disorders is the accumulation of misfolded protein aggregates, framing them within the classification concept of proteinopathies or “protein conformational disorders”. However, it is important to underscore that not all neurodegenerative diseases can be considered as protein conformational disorders [2].

Although the molecular underpinnings of neurodegeneration are still not completely understood, the increased oxidative stress associated with brain aging stands out as one of the common features across various neurodegenerative diseases [3,4,5,6]. Thus, aging is associated with increasing levels of pro-oxidant factors (reactive oxygen species, ROS) and the dysfunction of the antioxidant systems, leading to protein and cellular damage and ultimately to neurodegeneration.

The brain high vulnerability to oxidative stress appears to be mainly related to (i) its high oxygen demand (~20% of the body total basal oxygen), (ii) the high polyunsaturated fatty acids content of nerve cell membranes, (iii) the accumulation of transition metals, and (iv) the absence of efficient antioxidant systems [7].

In this line, several oxidation markers have been demonstrated to be elevated in several brain regions of patients with neurodegeneration, such as protein nitration (a marker of protein oxidation), 4-hidroxynonenal (4-HNE), and malondialdehyde (MDA) (markers of lipid peroxidation) [5,8].

ROS included several molecular species including superoxide (O2^•−^), peroxides (H_2_O_2_ and ROOH), and free radicals (HO^•^ and RO^•^), among others. ROS are generated by normal oxidative metabolism within the cell, with mitochondria being a major source; although other cellular components, such as endoplasmic reticulum-bound enzymes, cytoplasmic enzyme systems, and the surface of the plasma membrane, also contribute [9]. Therefore, a delicate balance between ROS production and the antioxidant defenses exists that protect cells. At regulated levels, ROS are involved in many vital physiological processes, such as cell differentiation and proliferation, apoptosis, and cytoskeletal regulation [10]. However, when this balance is progressively disturbed under certain conditions, such as aging, ROS species inflict damage to biomacromolecules with deleterious effects on proteins, lipids, and DNA, contributing to neurodegeneration.

About protein damage, both backbone and sidechains are susceptible to oxidative modification. Backbone oxidation causes protein cross-linking and peptide-bond cleavage, yielding products with lower molecular weight. On the other hand, sidechain modifications cause the loss of catalytic or structural function [11,12].

ROS are responsible for different lesions in both nuclear and mitochondrial DNA, including oxidized DNA bases, abasic sites, and DNA strand breaks, which ultimately lead to genomic instability [13,14]. ROS can also react with RNA; mRNA and rRNA molecules that are especially vulnerable to oxidative damage [15].

Cellular membranes and organelle membranes are especially susceptible to ROS damage, due to their high polyunsaturated fatty acids. ROS-induced lipid peroxidation exerts its deleterious effects through two general mechanisms: (i) loss of the integrity of cellular membranes (it is important to note that this process affects not only the cell membrane, but also the mitochondria, endoplasmic reticulum, and nuclear membranes), and (ii) generation of intermediate and end products, such as lipid hydroperoxides, MDA, 4-HNE, and acrolein that leads to genotoxicity, cytotoxicity, and ultimately, cellular death [16,17,18,19,20]. This type of regulated non-apoptotic peroxidation-driven and iron-dependent cell death mechanism is called “ferroptosis” [21]. Ferroptosis events have been established in neurodegenerative diseases [22,23,24,25], being considered as an alternative cell death mechanism in the absence of downstream indicators of apoptotic death [26].

Under physiological conditions, lipid peroxides are reduced to their corresponding lipid alcohols by glutathione peroxidases to prevent oxidative stress [27]. However, unfavorable conditions cause the accumulation of lipid peroxides, either for the inhibition of the reduction of lipid peroxides or for the increase of the production of lipid peroxides [28].

Glutathione peroxidase 4 (GPX4) prevents cells against ferroptosis reducing toxic lipid peroxides to nontoxic lipid alcohols, using glutathione as a cofactor [29,30]. Thus, the inactivation or absence of GPX4 causes the accumulation of lipid peroxides, leading to ferroptotic cell death [28,31].

Lipid peroxides are produced by three pathways: (i) through the Fenton reaction, (ii) by oxygenation and esterification of polyunsaturated fatty acids, and (iii) by lipid autoxidation. All these routes have in common that are iron-dependent either enzymatic or non-enzymatic manner [28]. Thus, high levels of iron trigger the increase of the production of lipid peroxides and in consequence, an overproduction of ROS.

Transition metals, such as iron, copper, and zinc, are necessary for brain functioning, participating in essential biological reactions as cofactors of metalloproteins that intervene in constitutive cellular functions, such as cell proliferation and differentiation, DNA synthesis and mitochondrial respiration, and in specialized cellular functions, such as, oxygen binding and transport, myelination, drug metabolism, and synthesis of neurotransmitters [32,33,34,35,36,37]. However, iron and copper are very efficient catalysts of ROS production. Thus, metal dyshomeostasis yields high levels of ROS, leading to oxidative stress and mitochondrial dysfunction [38,39,40,41]. In turn, mitochondrial dysfunction triggers a higher ROS production, driving to a “vicious cycle” inducing cellular damage. For this reason, transition metal levels require a tight control that tunes their levels and location.

In neurodegenerative disorders, increases in transition metal levels have been reported, being iron the metal showing the most substantial change [42,43,44]. In Alzheimer’s disease (AD), the accumulation of metals is observed within β-amyloid senile plaques [45,46,47,48], and alterations in iron, copper, and zinc levels have been found in several regions of AD brain [49,50,51,52]. Similarly, disturbances in metal levels have been reported in Parkinson’s disease (PD) [53,54,55,56,57,58,59]. The molecular mechanisms through which metal dysregulation triggers neurodegeneration are diverse, but they appear to be mediated mainly by the generation of free radicals and oxidative stress [4].

Ferritinophagy is the autophagic degradation of ferritin, the main iron storage cellular protein. Ferritinophagy could provide a link between autophagy impairment and iron dyshomeostasis in neurodegenerative disorders [60,61,62]. Thus, ferritin catabolism increases the availability of iron as a substrate for Fenton chemistry, which promotes ROS production and lipid peroxidation and leads ultimately to ferroptosis [63].

Among these noxious mechanisms, lipid peroxidation enhanced by iron appears to be a relevant triggering factor for neurodegeneration since the brain is highly enriched in polyunsaturated fatty acids and iron excess promotes ROS production.

Additionally, several pieces of evidence support a link between oxidative stress and protein aggregation, a feature that characterizes neurodegenerative proteinopathies, including AD and PD [64]. Transition metals, such as iron, copper, and zinc can be coordinated to Aβ peptide and α-synuclein, promoting their aggregation and leading to the formation of the histopathological hallmarks of AD and PD, respectively [65,66,67,68,69,70,71,72]. In PD, iron redox chemistry promotes the aggregation of α-synuclein, which is oxidation status-dependent. Thus, the addition of iron (II) in presence of oxygen induces the formation of α-synuclein oligomers, preventing its further conversion into β-sheet fibrils, a hallmark of PD, while, the addition of iron (III) triggers the formation of fibrils [73]. In AD, it was demonstrated that copper binds to Aβ peptide [74]. The resulting protein–metal complex aggregates are directly involved in ROS production, exacerbating the oxidative damage [67,75,76,77,78,79].

## 2. Melanins Synthesis and Properties

Melanins are a diverse group of natural substances ubiquitously found in nature that are responsible for various pigmentations present in most taxonomic kingdoms [80]. Their formation involves the oxidative polymerization of l-tyrosine, mainly in the skin, hair, and eyes of animals and many other phenolic compounds in plants and microorganisms [80]. In humans, melanins can be classified into three main types, depending on their composition: eumelanin (EU) and pheomelanin (PHEO) mainly found in skin melanocytes, and neuromelanin (NM) localized in some specific areas of the brain [81].

In the epidermis, both EU and PHEO are produced by the skin melanocytes inside specific organelles called “melanosomes”, which are the site for synthesis, storage, and transport of melanin. Melanin synthesis depends on the enzyme tyrosinase that carries out the oxidation of l-tyrosine [82] that through cyclation into 5,6-dihydroxyindole (DHI) and its carboxylated derivative 5,6-dihydroxyindole-2-carboxylic acid (DHICA) became the main building blocks of melanin, presenting several different active positions for polymerization [83,84,85,86]. Using voltammetric techniques and X-ray photoelectron spectroscopy, it has been studied the behavior of both units in redox conditions, specifically the effect of metal cations and ROS induced by hydrogen peroxide. DHICA subunits seem to be the responsible component of melanin to be capable of chelate metal ions, and so, to confer them highly antioxidant properties, becoming the most reactive part of the substance [87]. These features are due to hindered inter-unit conjugation triggering destabilizing effects that induce the formation of non-planar stacking. In this sense, the reactive groups are more exposed and may interact with ROS, metals, or other molecules. This arrangement also explains the high capacity for absorption in the 320 nm UV region, which makes of the DHICA subunits a great photoprotector. Otherwise, DHI subunits could be considered the structural element by its efficient inter-unit π-conjugation. As a result of this, DHI units tend to form 2D structures, stacked in onion-like nano-sheets, which complicates the interaction with other molecules [88,89].

The diversity of polymerization sites of the DHI and DHICA molecules and the incorporation of proteins and other molecules produce a substance that is highly heterogeneous in size and chemical composition precluding its fine structural characterization.

PHEO differs from EU by the presence of sulfur atoms incorporated through the binding of sulfhydryl groups of l-cysteine or other molecules such as glutathione to the 5,6-dihydroxyindoles [86,90], thus becoming benzothiazole molecules as the main polymerization subunits and conferring to the pheomelanins a yellowish or reddish color.

NM is a neuronal pigment that can be found mainly in brain regions enriched in dopaminergic or noradrenergic neurons, specifically in dopaminergic and noradrenergic neurons from *substantia nigra pars compacta* (SN) and *locus coeruleus* (LC), respectively; but also to a lesser extent in other regions of the central nervous system (CNS) and peripheral nervous system (PNS) [91,92]. According to its composition, NM could be considered as a mixture of EU and PHEO, being the pheomelanin components at the core of the NM granules and the eumelanin at the surface of the granules after the exhaustion of sulfhydryl-containing molecules [93,94,95]. However, opposite to EU and PHEO whose synthesis depends on the enzyme tyrosinase that carries out the initial oxidation of l-tyrosine, the synthesis of NM in the neurons is not enzymatically regulated [96], getting accumulated in organelles of autophagic nature [82,97,98,99]. Thus, its composition and structure are even less defined than EU and PHEO, and it seems to be composed of a mix of cysteinyl derivatives of oxidized dopamine (DA) or noradrenaline (NA) produced during the neurotransmitter synthesis in catecholaminergic neurons [100,101]. In dopaminergic neurons of the SN, the DA molecule is produced by the initial hydroxylation of l-tyrosine by the tyrosine hydroxylase to generate l-3,4 dihidroxifenilalanina (L-DOPA or levodopa) that is further converted into DA by the aromatic amino acid decarboxylase. In noradrenergic neurons of the LC, the DA molecule is further hydroxylated to produce NA by the action of the enzyme dopamine-β-hydroxylase.

Despite lacking enzymatic regulation, it has been observed a kind of balance between precursor monomers and NM that may be considered as a potential NM regulation mechanism. Thus, when there is an excess of cytosolic catecholamines that are not stored in synaptic vesicles, NM is produced [98]. In fact, there is evidence pointing to a reduced NM synthesis associated with the vesicular monoamine transporter-2 (VMAT-2) over-expression, while the opposite is found in those dopaminergic neurons with low expression of VMAT2 [98,102,103]. In in vitro experiments using PC12 cells as an NM cellular production model, it has been found that under high ROS levels, the tyrosine hydroxylase activity is increased, which may lead to increased intracellular DA derivatives. At the same time, as the DA reservoir increases, NM synthesis is upregulated in PC12 [104]. This feedback loop may be an indirect mechanism to cope with oxidative stress taking advantage of NM properties as an antioxidant.

Both DA and NA essential neurotransmitters involved in the regulation of brain states related to vigilance, action, reward, learning, and memory processes [105]. However, small amounts of these neuromodulators can be further oxidized by reactive oxygen species, yielding o-quinones that renders dihydroxyindole by cyclation and then 5,6-indolequinone or add l-cysteine that are able to polymerize and crosslink [80,106]. Thus, NM is constituted by the incorporation of both indole and benzothiazine molecules, together with lipid and proteins, producing a highly heterogeneous substance [102].

## 3. Function and Involvement of Neuromelanin in Disease

In general, neurodegenerative disorders appear to have a focal origin from which the pathology spreads. AD pathology expands from the cholinergic nuclei in the basal forebrain and the noradrenergic nuclei in the brainstem, most importantly the *Locus coeruleus*. PD brain pathology appears to initiate in the brainstem or limbic system with the greatest affectation of the *substantia nigra*. Interestingly, degeneration of cholinergic nucleus basalis is characteristic of AD, but also occurs in PD; while, neuronal loss in dopaminergic SN is the pathological hallmark of PD, but also occurs to a variable degree in AD [107,108]. Depletion of neurons in the noradrenergic LC is also recognized in both disorders, and for both AD and PD the greatest neuronal loss is found within the LC [109].

Despite that aging-related oxidative stress is a global process that affects all neurons, certain neuronal populations are selectively vulnerable in different neurodegenerative diseases, suggesting that additional factors play a role in the pathophysiological mechanisms of neurodegeneration of specific neurons and brain regions [110,111].

Dopaminergic and noradrenergic neurons in SN and the LC, respectively, are exposed to additional oxidative stress due to their inherent metabolism of catecholamines, the presence of highly unsaturated fatty acids, and iron (III) ions. Contrary to what would be desirable, the insufficient levels of glutathione, together with the lack of a mechanism to manage such significant amounts of ROS [112,113,114]. These neuronal populations are characterized by an age-dependent accumulation of NM [115]. It has been suggested that NM plays a protective role in these neurons by preventing the accumulation of catechol derivatives generated from DA or NA auto-oxidation, thus avoiding their toxic effect [102].

In humans, as well as in other primates, NM pigmentation of the brainstem can be detected macroscopically during the neuropathological analysis of the brain. Since the presence of NM in catecholaminergic neurons is associated with the catecholamine metabolism itself, the accumulation of NM in the soma of these neurons begins as early as the first decade of human life. NM levels gradually increase with age throughout life, until we reach the age of 60, approximately [116,117]. Afterward, the total amount of NM appears to stabilize due to the balance between the number of degenerating melanized neurons, and the ever-increasing NM levels in the surviving neurons, as confirmed by early histological studies in higher animals [117,118]. Thus, NM appears to play a role both in normal aging and neurodegeneration. However, it is important to underscore that many other adult mammals, including commonly used laboratory animals, such as mice and rats, do no present NM in their brainstem, even though these species have dopaminergic and noradrenergic neurons [81,106,119,120,121,122]. Thus, the relation between catecholamine metabolism and NM production is not quite direct and straightforward. An interesting approach to explore this relationship was recently described by Miquel Vila´s group that through the overexpression of human tyrosinase in the SN of rats, reported an age-dependent production of NM within dopaminergic neurons to the levels found in older humans [123]. They also found that the tyrosinase-overexpressing rats developed an age-dependent PD-like phenotype, including hypokinesia, Lewy body-like formation and nigrostriatal neurodegeneration [123]. All these data indicate that the NM plays a role both in normal aging and neurodegeneration, suggesting that it is necessary a well-balanced equilibrium between the catecholaminergic metabolism and the accumulation of NM.

A key feature of NM, as well as all other melanins, is related to their heterocyclic ring-structure with conjugated double bonds that creates a π-bond system across multiple atoms that lowers the energy and stabilizes the molecule. Due to this conformation, melanins have been attributed with many functionalities from which stands out its ability to trap UV–light, free radicals, toxins, and metals. Therefore, melanins constitute a primary protector barrier widespread in nature [116].

Historically brain NM has been considered a waste product of catecholaminergic neurotransmitter synthesis. NM was also thought of as a safe DA reservoir, avoiding the side effect of the highly oxidative catecholamines metabolism [121]. However, recent evidence suggests new roles for NM ranging from neuroprotection to neurodegeneration [81,121]. It has been suggested that the vulnerability of dopaminergic and noradrenergic neurons in SN and the LC, respectively, is related to their pigment content [124].

In normal conditions, NM production appears to play a protective role within the cell by preventing the accumulation of toxic catechol derivatives produced via the catecholamine auto-oxidation pathway [98,102,106], preventing ROS generation. NM presents also a chemoprotective role by interacting with a variety of potentially damaging molecules such as pesticides and neuroleptics [125], as well as potentially toxic cations such as iron, zinc, copper, manganese, chromium, cobalt, mercury, lead, and cadmium [101,126], acting like a “black hole” capable of chelating redox-active metals, especially iron [99,102,127]. Hence, NM is the main iron storage substance in dopaminergic neurons in the SN [128,129]. Interestingly, NM possesses two types of iron-binding sites (high and low-affinity sites); and under physiological conditions, iron binds preferentially to NM by the high-affinity site, avoiding its participation in ROS production and protecting cells from oxidative stress. However, this protective function appears to be lost at high iron concentrations, where the high binding site is saturated and the remaining iron binds to the NM by the low-affinity site. Thus, NM becomes a pro-oxidative burden, where iron could easily be released to participate in oxidative processes [130]. Therefore, the NM anti-oxidant or pro-oxidant role will greatly depend on the result of the regulation of iron/NM ratio [99].

Interestingly, besides NM, melanin is present in degenerative processes of nerve cells involved in sight and hearing in the retina and the inner ear, respectively.

In the eye, the outermost layer of the retina is a pigmented epithelium, called “retinal pigment epithelium” (RPE). The RPE is composed of a single layer of hexagonal cells that nourish the adjacent retinal visual cells, named “cones” and “rods”. Cones and rods are the photoreceptor cells in charge of photopic vision (vision under high light conditions) and scotopic vision (vision under low light conditions), respectively. Both types of cells are glutamatergic neurons sharing a similar structure. The outer parts of both cones and rods are formed by disks containing a photosensitive molecule called “opsin” being in close contact with the RPE.

The RPE owes its pigmentation to the presence of melanosomes containing melanin that is synthesized by the same pathway as the one produced by skin melanocytes. Densely packed melanin granules absorb light, thus reducing light back-reflection onto the retina. Changes in RPE melanin content have been described to be associated with normal aging, as well as in diseases such as albinism and age-related macular degeneration (AMD) [131].

AMD is caused by the progressive death of the photoreceptor cells within the macula of the retina, which is located next to the optic nerve, leading to a gradual loss of vision, with onset usually after the fifth or sixth decade of life. Non-exudative AMD, the most common form of the disease, is the main cause of blindness in older people. Mounting evidence indicates that the highly oxidative environment of the RPE, which is directly related to its high metabolic requirements, plays a major role in the pathogenesis of non-exudative AMD [132]. Intriguingly, AMD shares several epidemiological, clinical, and pathological features with other neurodegenerative disorders such as AD and PD. All of them are age-related, late-onset disorders, pathologically characterized by the presence of detrimental intra- and extracellular deposits, and oxidative stress and inflammation appear to play a crucial pathophysiological role. Moreover, hypercholesterolemia, hypertension, obesity, arteriosclerosis are established risk factors for the development of both AMD and AD [133,134,135,136].

In the inner ear, recent evidence points out that the presence of a population of melanocytes is vital for the proper functioning of sensory cells and hearing. These melanocytes are contained within the highly vascularized area of the cochlea, called “*stria vascularis*”, which appears to be involved in the chemical modulation of sound signal transduction [137].

Interestingly, Gi and collaborators [138] observed that, in a model of progressive hearing loss mediated by Vitamin A deficiency (VAD), the progressive hearing loss could be prevented by the activation of cochlear strial melanocytes, suggesting a protective role for the melanocytes through its antioxidant and free-radical scavenging properties. Moreover, they found a feedback loop mechanism by which VAD induced melanin production within the melanocytes by the upregulation of the expression of tyrosinase, the rate-limiting enzyme in the synthesis of melanin [138]. In the same line, in an albino mouse model, Murillo-Cuesta and collaborators [139] demonstrated that hearing loss was associated with the absence of cochlear melanin or its precursor metabolites, suggesting that melanin and its precursors, such as L-DOPA, have a protective role in the mammalian cochlea in age-related and noise-induced hearing loss.

Previous studies indicated a positive association between numbers of melanocytes in the skin and the human inner ear [140,141]. More recently, an evolutionary study has generalized these findings, suggesting that the levels of skin melanin correlate with internal (nonskin) melanin [142,143]. Several studies indicate that darker skin pigmentations are associated with a lower prevalence of hearing loss [144,145,146,147], thus suggesting again a protective role of melanin in hearing loss processes.

## 4. Crosstalk between Neuromelanin and Lipofuscin in Lipid Peroxidation

Along with the increase in NM during normal aging, another striking change throughout the brain is associated with the progressive accumulation of lipofuscin (LF) aggregates, which is traditionally considered as an “aging pigment” [148,149].

It is widely accepted that LF originates from lipid and lipoprotein peroxidation like an undigested bioproduct of central cellular detoxification processes of autophagy. It is associated with both normal cell-aging and neurodegeneration, reflecting the impairment of the exocytosis and lysosomal secretion systems [97,150]. Interestingly, the main component of LF comes from highly reactive lipid derivatives such as 4-HNE that form adducts by reacting with histidine, lysine, and cysteine residues, which also block proteasomal activity. Both NM and LF are poorly degraded, being accumulated throughout life in vacuoles of autophagic nature, potentially altering normal cellular degradation and secretory processes. The slowdown of the detoxification pathway has dramatic consequences in postmitotic neurons, where the metabolic “garbage” continues to accumulate during the entire cell life.

A possible link between metabolic production of NM and LF has been early pointed out in some pathological processes [151,152,153] such as some cutaneous melanosomes, where NM is shown as LF-melanin granules. Van Woert [154], using spectroscopic techniques in pigments of the SN, pointed out NM as melanin without LF. The colocalization of both types of substance could be indicating the production of a redox crosstalk between NM and LF, where the highly pro-oxidant effect of transition metals such as iron may be quenched by NM or amplified by a lipid peroxidation cascade initiated by LF lipids. Goldfisher indicated early on a potential lysosomal origin of LF in association with metals, indicating also that the presence of melanin was associated with catechol derivatives in the presence or absence of LF or its precursors, thus offering different mechanisms for melanogenesis [153].

Considering the NM formation as a not enzymatically-controlled auto-oxidation process, the presence of pro-oxidative LF is related to an increased oxidative state that could drive NM formation [151,152,153]. Barden [155] demonstrated by histochemistry and in situ enzymatic activity determination (light and dark-field microscopy, and UV light microscopy) that in the presence of ferrous sulfide, LF could be transformed into an NM-like substance after a pseudoperoxidation process, becoming melanized in the presence of DOPA or its precursors and showing similar properties to NM.

The same process was observed in the rare neurodegenerative Hallervorden-Spatz syndrome involving the basal ganglia, where both LF and NM are present in large swellings, named “spheroids”, found in the proximal portion of neurodegenerating neurons of the SN [156]. The authors suggested that LF arises from the lipid peroxidation of myelin fatty acids and other lipids that occurred in the presence of iron [156]. The fact that NM could evolve, at least in part, from the conversion of LF through a process of peroxidation in the presence of catecholamines suggests that NM is also linked to the metabolism of lipids. The potential conversion of LF into NM-like substances mediated by a dynamic phenomenon pseudoperoxidation of lipids supports the colocalization of both age-related pigments observed in different brain areas [101,157], and other associated tissues as the choroid vascular epithelium and RPE [158].

Even though the physiological roles of both pigments in the human brain are controversial, the lipid component and the presence of metal ions seem to be a common pattern to NM and LF that will determine their antioxidant or pro-oxidant role [159]. Thus, the analysis of the balance between NM and LF could be relevant for a better comprehension of normal aging and neurodegenerative processes.

## 5. Neuromelanin and Immune Response

With aging, there is a global low-level increase in the brain inflammatory response. However, in neurodegenerative diseases, there is also a strong increase in the inflammatory response in specific and restricted areas of the brain, where extracellular cell-released NM could be playing a relevant role in triggering and sustaining the inflammation process [160,161,162,163]. Regardless of the cell death mechanism, once a catecholaminergic neuron dies, the NM, loaded with all the substances that were bound while in the cell, is released to the brain parenchyma. Each component bound to the biopolymer might be released or not; but depending on their nature, they may contribute to initiate or exacerbate an immunological response (Figure 1).

### 5.1. Neuromelanin and the Innate Immunity

Microglia, as brain-specific resident macrophages, can engulf the released NM to clear it (see Figure 1, steps 1 and 2). In this line, Zhang and colleagues showed that human NM can be phagocytized by primary enriched microglia Fisher 334 rat cell culture [164]. Its degradation seems to be mediated by ROS production by the phagocyte oxidase (PHOX) enzyme, which is responsible for a respiratory burst. In addition, Depboylu et al. [165] demonstrated through immunohistochemistry and in situ hybridization techniques that in the SN and other brain regions from PD patients, there is an increased expression of C1q in microglia cells surrounding extracellular coming from dead NM-containing neurons, thus facilitating opsonization and phagocytosis by microglia. Interestingly, the same analogous process, mediated by C1q, appears to be relevant for the clearance of Aβ peptide plaques by the microglia (see Figure 1, step 2) [165,166]. Recently, an important role is being attributed to C1q in neurodegeneration [167] Here, we propose that the competition between the clearance of NM and amyloid plaques by C1q opsonization and microglia phagocytosis may result in the blockage of the physiological system of clearance, leading to the accumulation of deleterious components and the progression of the neurodegenerative process.

Another relevant pathway for NM to activate the innate immunity mechanisms of microglia is through Pattern Recognition Receptors (PRR). Since NM has a repetitive structure, it is a good candidate for interfering with this pathway. Extracellular NM can activate in vitro microglia towards the M1 proinflammatory profile by induction of pattern-recognition receptors such as toll-like receptor 2 (TLR2) and nucleotide-binding oligomerization domain 2 (NOD2) (see Figure 1, step 2a) [168]. Extracellular NM fragments bind to TLR, also expressed in astroglial cells, to induce the NF-κB expression by the MyD88 transduction signaling, which is a known pathway to start inflammation by releasing proinflammatory cytokines, such as IL-1β, IL-6, and TNFα (see Figure 1, step 4) [168]. By the same pathway, Aβ peptide brain deposits seem to be cleared by TLR2, lessening cognitive decline in AD and aging [162,169]. Once again, these results point to a competition for the resources and mechanisms capable of removing amyloid plaques and extracellular NM deposits. 

Microglia over-activated by extracellular NM secretes large quantities of proinflammatory cytokines (Figure 1, step 3); which in turn, induce the expression of NO synthase (iNOS) and the production of NO, further increasing free ROS. NO is associated with neurodegeneration by different mechanisms: (i) iron dyshomeostasis, (iron imbalance is involved in both AD and PD) [170], (ii) lipoperoxidation in association with transition metals [171], and (iii) antigenic modifications due to ROS derivates of NO [172,173]. Together with the intrinsic effect of ROS production, these NO-related mechanisms represent a strong inflammatory stimulus that can mount a sterile immune response involving other cells as astroglia and T lymphocytes and the production of other proinflammatory cytokines (Figure 1, step 5).

### 5.2. Neuromelanin and Acquired Immunity

In this context, two relevant findings point to the activation of microglia by NM as a potential triggering factor of the autoimmune response involved in PD. Orr and colleagues studied a humoral mechanism capable of causing microglial-meditated injury in tissue from idiopathic and genetic PD compared to controls [174]. They detected IgG immunoglobulins recognizing dopaminergic neuronal structures that correlated with pigmented neurons. Moreover, they found MHC class II immunopositive microglia expressing the high-affinity IgG receptor FcγRI, which is consistent with a potential phagocytic attack on the IgG-immunopositive pigmented neurons (Figure 1, step 5). In addition, the targeting of pigmented neurons by antibodies may also trigger antibody-mediated cytotoxicity. In line with this, Double et al. found that antibodies against melanin-skeleton were increased in patients with a clinical diagnosis of PD [175]. Moreover, the concentration of antibodies detected correlated negatively with the disease duration, while no correlation was found with the severity of the disease.

Thus, it seems that there is a first innate response, which induces a beneficial inflammation to help to clear the melanin remnants, evolves into a more complex and difficult-to-manage pernicious immune response mediated by cellular immunity.

Microglia, as a macrophage-related cell, expresses MHC class II and it is the main cell population able to present antigens in the CNS [176]. Nevertheless, under a pro-inflammatory environment, several cells inside the CNS can act as Antigen-Presenting Cells (APC), such as astrocytes, border-associated macrophages (BAM, which reside in the choroid plexus, the pia, and the dura maters), and dendritic cells (DC) in the blood–brain barrier [177]. In this line, Oberländer et al. used human NM to stimulate murine DC, and observed how the DC can capture the NM and be activated as APC expressing MHC class II and CD86 in their cell surface [178]. Thus, they propose microglia as the initial APC that triggers NM-driven autoimmune-based pathogenesis of PD.

The phagocytosis of NM by APC cells opens a new potential avenue with immunological consequences [179]. Extracellular NM and their derived components would be presented by MHC class II, leading to two different pathways: (i) activating B cells to produce autoantibodies, and (ii) cytotoxic response priming naïve T cells, which under a pro-inflammatory environment mediated by cytokines that could infiltrate from subarachnoid vessels (see Figure 1, step 5a) [180].

Despite the immune privilege of the CNS, the priming process is possible because the CSF containing immune cells and brain antigens is transported by brain lymphatic vessels to the deep cervical lymph nodes (dCLN), where the APC can present all those components attached to NM as brain antigens to naïve T cells [165]. Upon their activation in the context of neuroinflammation, they can be chemoattracted and mobilized towards the CNS [181]. At the same time, APC would prime B cells to turn into plasmatic cells to secrete IgGs, thus explaining the presence of IgG against NM in sera from PD patients [175].

As NM and their components are presented by MHC class II and activate an immune response, it is also plausible that in the proper environment (e.g., presence of IFNγ), they may be presented by MHC class I (Figure 1, step 5b). TLR, proinflammatory cytokines such as TNFα, or even aging [181] can induce astrocytes to become reactive astrocytes, leading to the secretion of IFNγ, which promotes MHC class I expression in catecholaminergic neurons. Then, catecholaminergic cells may present NM portions together with protein oligomers or toxic components. Previously, these proteins or peptides may have experienced post-translational changes enhancing their antigenicity, and thus triggering an autoimmune response that targets both degenerating and healthy dopaminergic neurons for cytotoxic T lymphocytes (Tc) [182]. The infiltration of activated Tc cells, exacerbation of the neuroinflammation, and loss of neurons may contribute to the progression of pathology in different neurodegenerative diseases.

In this situation, the CXCL10 pathway is another checkpoint where NM could interfere. CXCL10 is a chemokine strongly secreted in response to interferon, as well as by the expression of NF-κB induced by TNFα. CXCL10 appears to function as a chemo-attractive for activated T cells, monocytes/macrophages, and microglia, promoting antimicrobial activity, and inducing astrocyte proliferation. Although its role is not well understood, depending on the context may be neuroprotective or neurotoxic. CXCL10 levels correlate with both β-amyloid [183] plaques and mini-mental state examination (MMSE) [184], but it is still not clear whether it is increased in the CSF of AD patients versus controls [185,186]. In the PD murine model induced by 1-methyl 4-phenyl 1,2,3.6-tetrahydropyridine (MPTP), CXCL10 mRNA expression was up-regulated in the striatum and the ventral midbrain [187], and it also seems to be related to HIV-1 dementia and Tick-borne encephalitis neuropathology [186]. Tousi and colleagues [188] showed in vitro how NM in the presence of TNFα inhibits CXCL10 expression by inhibition of NF-κB activation in the human A172 astroglial cell line. Nevertheless, no effect was detected when NM was present alone without a proinflammatory background. In the context of CXCL10 neurotoxicity, NM appears to be neuroprotective. Moreover, the fact that NM is able of inhibiting the expression of CXCL10 in an NF-κB-dependent manner may have wider implications, since the same pathway regulates many pro-inflammatory cytokines at the same time.

## 6. Conclusions

As previously discussed, the accumulation of NM is a process that is not enzymatically regulated. With aging, the cell appears to be unable to balance adequately between the production of NM that helps to cope with increasing oxidative stress, and toxicants binding on one side, and the production of catecholamines on the other side. Thus, the aging cell cannot keep increasing NM production at the expense of a reduced production of neurotransmitters that would lead to neurodegeneration associated with a loss of function of the cell.

In normal physiological states, intracellular NM plays a protective role for the neuron as a metal chelator, preventing ROS production and therefore oxidative damage. Eventually, NM could even act as an immunomodulator through NF-κB inhibition, which regulates proinflammatory cytokine expression. However, when the concentration of active metals and other toxicants surpasses a certain threshold, the antioxidant potential of NM turns into a pro-oxidant role, in a feedback loop, producing an increased cellular oxidative stress.

In addition to PD and AD, AMD and certain neurodegenerative hearing disorders involve the degeneration of nerve cells in the context of a melanized tissue. However, opposite to the NM present in the catecholaminergic neurons of the SN and the LC, the melanin in the tissue affected by AMD and degenerative hearing disorders is not present in the neural cells themselves. In these cases, it has been suggested that the presence of melanin in a cellular context with high metabolic requirements helps the cell to cope with oxidative stress. In line with previous findings on brain NM, these findings suggest that the neurodegenerative process in catecholaminergic neurons is initiated by events not directly related to the presence of NM, but rather to the oxidative metabolism of the catecholamines, which is linked to oxidative stress and the generation of endogenous neurotoxins.

Oxidative damage includes, among others, lipid peroxidation that compromises cellular integrity and promotes the production of noxious products. Both processes lead to cellular death and the consequent release of NM from catecholaminergic neurons into the brain parenchyma. Thus, the release of NM, loaded with all the substances that were bound while in the cell, may contribute to initiate or exacerbate the strong brain inflammatory status observed in neurodegenerative processes, mediated by both an autoimmune and cytotoxic response that result in cell death and further NM release, closing a vicious circle that leads to the progression of neurodegeneration. Parallelly, the physiological mechanism of NM clearance appears to be like Aβ plaques, which relies on the opsonization by C1q and microglial phagocytosis. Therefore, in the context of AD, elevated amounts of Aβ peptide could hinder NM clearance and vice versa. As a result of the inability of the brain milieu to properly degrade both Aβ peptide and NM, they will remain more time lingering in the parenchyma, increasing neuroinflammation and tissular damage. Thus, the paradox of NM is that its greatest virtue is its worst defect. NM can play either a cytoprotective or a neurotoxic role depending on its cellular and extracellular context and the load that carries.

## Figures and Tables

**Figure 1 antioxidants-10-00124-f001:**
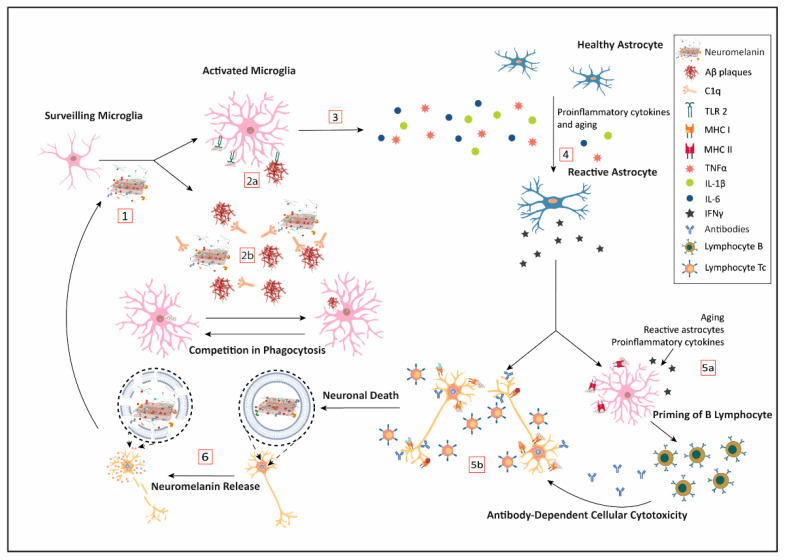
Immune response to the neuromelanin (NM) release to brain parenchyma. Upon neuronal cell death, surveilling microglia engulf cellular debris. The NM released to the brain parenchyma is less stable than the intracellular vesicle-contained NM, and therefore may distribute its toxic load throughout the tissue, triggering microglial activation (1). At the same time, NM may interact with the protein C1q which facilitates its phagocytosis, hindering, in turn, the clearance of amyloid deposits by competition (2b). Simultaneously, NM structure could interact with microglial toll-like receptor 2 (TLR2) (2a) promoting the secretion of pro-inflammatory interleukins (3). These secreted interleukins will amplify neuroinflammation, impairing other neurological structures, and transforming astroglial cells into reactive astrocytes that produce INFγ (4). Activated microglia by INFγ stimulus will express major histocompatibility complex (MHC) class I, displaying portions of NM and its load that in the deep cervical lymph nodes could prime B cells, potentially inducing an autoimmune response (5a). Parallelly, INFγ also induces the expression of MHC class I in healthy neurons, and similar to MHC class II in microglia, MHC class I can present modified antigens through its binding to NM, thus triggering Tc lymphocytes attack (5b). Finally, increased cell death is associated with a higher release of NM, establishing a feedback loop that exacerbates a pathological immune response (6).

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
