# Peer review of "The Neuromelanin Paradox and Its Dual Role in Oxidative Stress and Neurodegeneration"

_antioxidants, 2021, doi:10.3390/antiox10010124_

Round 1
Reviewer 1 Report
This is an interesting review. I have recommendations to adjust several mistakes in citations and few statements in order to improve this review.
Abstract. It should be xenobiotics not antibiotics
Line 64. These citations do not deal with Metals induced oxidative stress, and neurodegeneration, then they should be removed:
Dalle-Donne, I.; Trends Mol. Med. Apr 2003
Dalle-Donne, I.;. Clin. Chim. Acta Int. J. Clin. Chem. Mar 2003b
Line 69. Similarly, disturbances in metal levels have been reported in Parkinson's disease (PD)[32–35].
Authors should cite also these studies: Connor JR et al J Neurochem 1995; Riederer P et al J Neurochem 1989;
Iron accumulation in different neurodegenerative diseases like AD, PD and MS and has been imaged with MRI as discussed in an interestin review by Moeller H et al TINS 2019
Line 105. Here an original study showing the accumulation of NM in dopamine neurons of SN and noradrenaline neurons of LC should be quoted: Zecca L et al PNAS 2004
Line 110. …and peripheral nervous system (PNS) [63][67]. Neuromelanins are a family of compounds made of melanic, lipid and protein components and they occur in different brain regions. They are always bound to metals as described in these studies that should be cited (Engelen M PlosOne 2012; Zecca L PNAS 2008)
Line 113. ''NM belongs to the family of melanins, a big group of biopolymers ubiquitous in nature[68]. Although there is no consensus about its structure, it appears to be formed by a heme-like structure in a π-conjugated 112 matrix''. These sentences are partially wrong and should be rewritten. Neuromelanins have structure quite different from classical melanins (skin, hair melanins) and do not contain the π-conjugated system.
Line 126. …accumulated in organelles of autophagic nature[67,72,73]. This issue has been well discussed in Zucca FA Progr Neurobiol 2017
Line 128. …noradrenalin produced during the neurotransmitter synthesis[74,75]. The citation Plum, S. Sci. Rep. 2016 has nothing to do with NM strcture and should be deleted. Instead these should be cited: Wakamatsu K, et al J Neurochem. 2015;
Line 132. ...neurodegeneration[76]. here cite also : Zucca FA, Neurotox Res. 2014
Line 136. …oxidation pathway[66,78]. These two citations are two reviews on various aspects of NM. Cite also the original study: Sulzer D et al PNAS 2000
Line 148. …on the result of the regulation of iron/NM ratio[3].. The citation has little to deal with the statement: Barnham, K.J. Nat. Rev. Drug 345 Discov. 2004. This reference is more appropriate: Zucca FA Progr Neurobiol 2017
Line 163. Besides, since the formation of NM is a non-enzymatically controlled auto-oxidation process, the presence of metals and LF may produce and increased oxidative state that drives NM formation [90–164 92]. This sentence is not clear. Please rephrase it.
Line 180. The lipid component and the presence…This discussion on interaction between LF and NM and, oxidation of LF is based on old studies using mainly histochemical methods. Today studies with EM, immuno EM, isolation of NM, chemical spectroscopic and other methods have shown new aspects of this issue (Zucca et al npj Parkinson Dis 2018; Engelen et al PosOne 2012; Biesemeier A J Neurochem 2016; Ferrari E, et al ACS Chem Neurosci. 2017 ).Then authors should be careful with their statements
Line 188. …where extracellular cell-released NM could be playing a relevant role in triggering and sustaining the inflammation process[98]. This was first shown by McGeer, P.L Neurology 1988; Langston, J.W . Ann. Neurol. 1999.
Line 189. …role in triggering and sustaining the inflammation process[98]. The cited paper by Ouchi Ann Neurol 2005 is a PET study and is not related to release of NM and consequent nuroinflammation and should be removed. Instead this was suggested by Wilms H et al FASEB J. 2003; Zecca L et al Acta Neuropathol. 2008.
In fig 1 authors should show which is the drawing of NM.
Line 221. Here, we propose that…Authors should comment that NM and amyloid have similar characteristics and can play either protective or toxic role as discussed in Rao KS et al Prog Neurobiol. 2006.
Line 233. By the same pathway, Aβ brain deposits seem to be cleared by TLR2, lessening cognitive decline in AD and aging[104]. Here cite also Wilms H et al FASEB J. 2003, first showing microglia activation by NM mediated by NF-κB and release of tumor necrosis factor alpha, interleukin-6, and nitric oxide.
Reviewer 2 Report
The review paper by M. Calero and coworkers summarized the current knowledge about the role played by neuromelanin in aging and neurodegenerative diseases. In particular, the paper deals with the protective or pro-oxidant role exerted by the pigment in normal development and in neuronal disease conditions, respectively. The paper covers these aspects in satisfactory detail and gives an useful overview of the current understanding of the connection between neuromelanin toxicity and the progress of neurodegenerative pathologies, primarily Parkinson’s disease. I think that the review will be of interest to the readership of Antioxidants, but the Authors are invited to revise the paper taking into account the following comments.
- I noticed that most of the references cited in the paper refer to old literature, which is not useful for the readers willing to gain an updated overview of the field. For instance, taking into account the first 15 references, quoted in the session aimed at introducing the topic, only one is from 2018 and two from 2013, whereas all the others refer to papers published before the previous decade.
- Neuromelanin structure, page 3, lines 111-115. Neuromelanin can be hardly classified together with peripheral melanins, as both its structure and composition are markedly different. Unlike peripheral melanins, neuromelanin contains proteins and lipids covalently bound to the melanic component. In addition, the melanic component is not structured and in particular contains no stackings of pi-conjugated ring structures, which is the common arrangement of tissue melanins. This must be corrected before publication of the paper. A summary of the current knowledge of the structural features and composition of neuromelanin is actually given in ref. 81 and I am surprised that the Authors neglected this aspect.
- Minor points: Page 4, line 173, please correct “oxygen peroxide” with “hydrogen peroxide”; line 183, please note that neuromelanin and lipofuscin are better defined as “substances” rather than “molecules”.
Reviewer 3 Report
The review is topical and brings together relevant literature about the probable roles of neuromelanin in neurodegeneration. However, some drawbacks in design of the article hinders to gain insight in the issue.
- To my opinion, the major disadvantage of the review about neuromelanin in neurodegeneration is lack of a specific section about neuromelanin itself: what are the melanins, what is the composition of neuromelanin, it’s chemical and physiological properties, distribution in the CNS, ontogenetic and evolutionary (interspecies) differences in it’s distribution and amount. In fact, some of these data are presented, but are given irregularly, interspersed between facts on neurodegeneration, while, for better understanding of what is neuromelanin (the topic of the are view), a special separate section is needed. This section could to be placed after the description of neurodegeneration and it’s relation to metals and oxidative stress.
- Table 1 (line 330). The table header should be changed (expanded), since term “properties of neuromelanin” encounter all properties of the substance, including chemical and physical properties, while you give only physiologically protective and toxic ones.
2a. Being in the section Conclusions, Table 1 should summarize general conclusions made after discussion of the presented data, so that, any references should be avoided (all of them should have been presented in the previous text).
2b. In “neuroprotective” graph of the Table, you claim that neuromelanin is a potential dopamine reservoir. This hypothetical and, furthermore, controversial opinion should be discussed in appropriate section with appropriate references in the text of the review and only after such a discussion can be included in the Table with concluding results.
- line 140 and 183.
For both neuromelanin and lipofuscin is hardly applicable the term “molecule”. These are amorphous supramolecular complexes with very irregular and very complicated content. Therefore, they can be called substances, or, as a last resort, supramolecular complexes (supramolecules).
- lines 166 and 182:
The use of terms “reciprocal” interconnection and “interconvertibility” between neuromelanin and lipofuscin seems unfounded as you provide some data only that melanin can be formed from lipofuscin, but not vice versa.
There are some incorrect words and phrases in the text.
Line 34:
“neurons and synapsis”
line 108:
“NM is a dark neuronal pigment, therefore it can be mainly found in brain regions enriched in dopaminergic or noradrenergic neurons …”
Do you wish to say that NM is located in regions with dopaminergic or noradrenergic neurons because it is a dark pigment?
lines 119-121:
“Afterwards, the total amount of NM appears to stabilize due to the balance between the number of neurodegenerating melanized neurons, and the ever-increasing NM levels in the surviving neurons”
Either neurodegeneration, or degeneration of neurons.
lines 328-329:
“NM can play both a cytoprotective or neurotoxic role”
